# Exploring Gender Differences in the Instructor Presence Effect in Video Lectures: An Eye-Tracking Study

**DOI:** 10.3390/brainsci12070946

**Published:** 2022-07-19

**Authors:** Yuyang Zhang, Jing Yang

**Affiliations:** 1Bilingual Cognition and Development Lab, Center for Linguistics and Applied Linguistics, Guangdong University of Foreign Studies, Guangzhou 510420, China; 20200110079@gdufs.edu.cn; 2School of International Studies, Zhejiang University, Hangzhou 310058, China

**Keywords:** video lecture, instructor presence, gender differences, social presence, eye-tracking

## Abstract

The instructor’s presence on the screen has become a popular feature in the video lectures of online learning and has drawn increasing research interest. Studies on the instructor presence effect of video lectures mainly focused on the features of the instructor, and few have taken learners’ differences, such as gender, into consideration. The current study examined whether male and female learners differed in their learning performance and eye movement features when learning video lectures with and without the instructor’s presence. All participants (N = 64) were asked to watch three different types of video lectures: audio-video without instructor presence (AV), picture-video with instructor presence (PV), and video-video with instructor presence (VV). They watched nine videos, three of each condition, and completed a reading comprehension test after each video. Their eye movement data were simultaneously collected when they watched these videos. Results showed that learners gained better outcomes after watching the videos with a talking instructor (VV) than those with the instructor’s picture (PV) or without the instructor (AV). This finding suggests that the dynamic presence of the instructor in video lectures could enhance learning through increased social presence and agency. Gender differences were found in their attention allocation, but not behavioral learning performance. When watching the videos with a talking instructor (VV), female learners dwelt longer on the instructor, while males transited more between the instructor and the text. Our results highlight the value of instructor presence in video lectures and call for more comprehensive explorations of gender differences in online learning outcomes and attention distribution.

## 1. Introduction

Online learning is popular and widespread, especially during the COVID-19 pandemic when many schools experienced lockdown. Online learning, often in video lectures, can provide access to high-quality multimedia education resources without time and space constraints. However, it lacks face-to-face interactions between the instructor and learners, who may feel disconnected and less engaged in online courses. Do learners improve when an on-screen instructor is present? How could the instructor presence promote online learning?

According to the personalization principle of the social agency theory [1], the instructor on-screen presence in the multimedia instructional message, as social cues (such as eye gaze, facial expressions, body orientation, and gestures), could fuel a social response in the students and create social presence, a sense of partnership between the students and the instructor. Students try harder to make sense of the presented learning materials when they feel they are in a social partnership with the instructor. Thus, their increased interest and motivational commitment could lead to deeper cognitive processing of the learning materials and better learning performance [1,2]. However, the instructor’s physical image on the screen (such as a one-shot talking head or a picture of a cartoon character) does not substantially improve the students’ learning outcome, according to the image principle [1]. As the social agent, the on-screen instructor needs to engage in real human-like gestures to facilitate the learners’ interest, motivation, engagement, and learning performance.

In contrast, the cognitive load theory [3,4] regards the instructor’s presence in video lectures as a source of interference. Continual access to the instructor’s face and gestures during the lecture may divert learners’ limited attention from the learning content and create split attention between the instructor and the learning materials on the screen [5]. Frequent engagement switches between the instructor and the learning content might also overload learners. Their limited working memory capacity has to be devoted to additional extraneous processing that is not directly related to the instructional objective [4]. Thus, the interference effect caused by the instructor’s presence in the videos might offset the advantages of social presence it brings and even hampers learning at the worst.

Many empirical studies have examined the impact of instructor presence on students’ learning outcomes in video lectures, and the results are mixed. Some studies comparing students’ learning performance in instructor-present and instructor-absent video lectures support a significant role of instructor presence in students’ improved learning performance [6,7,8,9,10] and enhanced positive affective responses, i.e., learning satisfaction and situational interest [11,12]. The instructor’s facial expression, eye gaze, body orientation, gestures, and sizes have been testified to have various consequences on learners’ learning performance and attention allocation [13,14,15,16,17]. However, some researchers claimed that the instructor’s presence might capture students’ attention to the learning materials and impose a higher cognitive load on the students [18,19,20].

According to Mayer [21], the instructor presence effect can be subjected to boundary conditions, including the instructional content, instructional context, and individual differences. Many controversies might arise from learners’ differences, which have not been closely examined. For example, gender differences have been reported in the perceptions of social presence during e-learning, with females experiencing stronger perceptions of social presence than males [22,23]. However, little is known about gender-based sensitivity to instructor presence during video lectures, which is essential to understand individual differences in online learning outcomes. Considering the relevant literature involved primarily female participants, investigating gender differences in the effect of instructor presence in video lectures is necessary.

### 1.1. Instructor Presence Effect: Now You See It, Now You Do Not

The primary debate in this line of research lies in the facilitation effect of instructor presence in video lectures. For example, Kizilcec and his colleagues [24] revealed that 75% of students in their study preferred to learn video lectures with an instructor’s face. These students reported a better learning experience than those who did not see an instructor’s face in the video lectures. Another study found that students viewing videos with the instructor and PPT slides had better learning performance than those watching video podcasts with only PPT slides [25]. More recently, Hew and Lo [26] demonstrated that secondary school students had the highest scores in the recall and application questions in the video lectures with the teacher’s talking head. However, other studies using similar paradigms failed to find such an instructor presence effect [19,20,27,28]. For example, Homer and his colleagues [20] asked adult participants to view video lectures with the speaker or a no-video lecture with the audio and slides. They assessed the learning, cognitive load, and social presence in two groups of participants. Both groups did not differ in learning performance or social presence, but the video group experienced a greater cognitive load. Another study reported that video with instructor presence as a distractor impaired learning performance [28]. However, their learners preferred and believed this learning condition was most effective.

Hong and his colleague [29] provided a unique way to see the conflicting results. They revealed that instructor presence increased learners’ cognitive load when they learned procedural knowledge. Adding the instructor in a video lecture only facilitated declarative knowledge learning. In another study involving procedure knowledge learning, the authors tested the impact of teachers’ continuous vs. intermittent presence in instructional video lectures on procedural knowledge [30]. They found that a teacher’s intermittent presentational approach improved learning achievement and satisfaction and caused less cognitive load than the continuous presentation condition. So far, the inconsistent findings suggest that the on-screen instructor presence only plays an important role in some presenting modes. Fiorella and his colleagues [18] compared two instructional methods: a talking instructor with static diagrams or dynamically drawn diagrams without the instructor. Students were asked to adopt one of the learning strategies (explain, draw or rewatch). They found that the alignment of the instructional methods with learning strategies was important to the learning outcome instead of instructor presence.

Obviously, researchers are interested in the effectiveness of adding an instructor to the video lecture on learners’ learning outcomes. In this line of explorations, two main concerns are usually involved: instructional methods and learning outcomes. However, Mayer [21] suggested inserting a focus on the learning process between instructional methods and learning outcomes. Interviews, behavioral tests, and self-reports that have been mostly used could only make inferences about the information processing during online learning.

Eye-tracking technology is one of those measures that can shed light on the underlying attentional dynamics during learning. It has been adopted in some recent studies on instructor presence [8,9,10,11,12,14,19]. The eye–mind hypothesis postulates that the learner’s fixation and visual attention are linked [31]. The more fixation time is attributed to an item, the more visual attention is allocated to that stimuli. So far, the most commonly used eye-tracking measures in the relevant literature are fixation count and dwell time. More fixation counts and longer dwell time indicate more attention to an object/area. Using these measures, previous researchers showed that the onscreen instructor did divert some of the learners’ attention from the learning content: there was a shorter dwell time on the learning materials in instructor-present videos as compared to instructor-absent ones [7,9,10,19,27]. Apart from examining participants’ close attention to the instructor and other content on the screen, some studies also considered the participants’ number of transitions between the instructor and the content [11,12]. This measure has been regarded as an index for split attention caused by the instructor’s presence in the video. Learners have been shown to make more transitions between the instructor and content areas in instructor-present videos [11,12].

The increasing eye-tracking studies on the instructor presence effect have mainly focused on the impact of instructor presence on students’ attention allocation and learning. For example, Wang and Antonenko [11] made 26 participants view 10-min mathematics videos on easy and difficult topics with the instructor either present or absent. Although there were no significant group differences in their learning transfer, instructor presence improved recall for easy topics and decreased the self-reported mental effort for the difficult topic. In contrast, Pi and Hong [9] revealed that participants allocated more visual attention to the instructor than to the slides in a video podcast that a psychologist gave on attachment. The condition of the instructor talking and the slides led to the best learning performance. The video lectures’ topic might influence the instructor’s visual attention allocation.

To identify effective conditions that instructor presence work and the underlying visual attention process, many researchers examined social cues or features of instructor presence. These features include but are not limited to, the instructor’s eye gaze [14,15,19], facial expression [32,33], body orientation [14], gestures [34,35], image size [36], and position [17] on the screen. Researchers investigated the instructor’s eye gaze mostly to testify whether continual access to an instructor’s eye gaze can guide and improve learning. Van Gog and his colleagues [10] revealed that the face of the instructor in the problem-solving modeling video was beneficial to participants’ learning performance. van Wermeskerken and van Gog [19] compared similar demonstrating videos with instructor’s gaze guidance (i.e., staring straight into the camera) present or absent. They failed to find any facilitation or hinder effect of the instructor’s face or eye gaze on learning performance. Still, both affected the visual attention allocation when participants viewed the videos. In another study, students viewed organic chemistry video lectures with the instructor’s direct gaze (the instructor looked into the camera in a transparent blackboard context) or the instructor’s gaze guidance (the instructor looked and wrote on the blackboard) [37]. The two groups did not differ in learning performance and engagement. Finally, Pi and her colleagues [14] extended the studies on eye gaze. They examined the effect of the instructor’s eye gaze and body orientation on attention allocation and learning in video lectures. Their learners who viewed the instructor’s guided gaze paid more visual attention to the slides, while learners who viewed the instructor’ s direct gaze spent more attention on her face. The former group had better retention and transfer outcomes. The body orientation did not play any significant role. These explorations mentioned above help answer the key question that educators and researchers care about: how to optimize the design of video lectures to improve students’ learning.

Compared with the efforts on the instructor features and learning materials, learners’ differences in the instructor presence effect have not been closely examined. In Kokoç et al. [8], participants of different sustained attention levels watched three types of video lectures (picture-in-picture, voiceover presentation, and screencast) which differed in instructor presence. At the same time, their eye movements were simultaneously recorded. Due to the heterogeneity in content and multimedia elements across different video types, they conducted separate analyses of the eye-tracking measurements for each type. Results demonstrated that only the picture-in-picture type with instructor presence resulted in different eye movement features between learners of high and low sustained attention levels.

Kokoç and his colleagues claimed that modeling individual differences in the design of video lectures is still at an early stage in the literature [8]. Learner characteristics matter as it is thought to be one of the most important issues to consider when designing effective e-learning environments [38,39]. Unfortunately, it remains unclear whether presenting an instructor on the video screen has the same effect on learners of different gender, age, and cognitive abilities.

### 1.2. Gender Differences in the Perceptions of Online Social Presence

According to the gender similarities hypothesis [40], males and females are similar on most, but not all, psychological variables. For example, meta-analyses have shown that gender differences have been reliably found in cognitive skills such as attention, memory, and spatial ability [41,42,43]. These cognitive differences could influence their processing and learning procedures [44]. For example, it was found that females with lower spatial ability benefited more from animated instructional presentations than males [45]. Thus, the same instructional interventions could impact the two groups differently [44,46].

Gender differences have been examined in online learning environments. One of the issues that interest researchers is whether males and females differ in their perceptions of social presence in e-learning settings. Online social presence refers to “the subjective feeling of being connected and together with others during computer-mediated communication” [47] (p. 1739). It has been assumed to be crucial to the success of online learning. Previous studies have demonstrated that as a positive experience, social presence positively influences online learners’ satisfaction [48,49,50] and performance [51,52]. However, the same e-learning environment can result in different subjective experiences of presence between male and female learners, with females having greater perceptions of social presence than males [22,23].

In a web-based introductory information systems course, Johnson reported that women communicated more, experienced higher social presence, and performed better than men [22]. Johnson attributed females’ stronger perceptions of social presence to the gender-related differences in communication as females were found more attuned to the socially oriented aspects of communication. Unlike Johnson, Rodríguez-Ardura and Meseguer-Artola attempted to explore the cognitive and emotional factors that contributed to social presence experience and considered the moderating role of gender [23]. They found gender moderated the relationship between emotion and presence, with women more sensitive to emotion in their presence formation than men. In other words, the greater the emotional effort women experience, the more intense their experience of presence. Though the relevant empirical studies have been limited, the potential differences between males and females indicated by the existing evidence still show the necessity for future presence-related research to take gender into account.

Suppose the instructor-present video lectures activate a higher level of social presence than those instructor-absent videos. In that case, the instructor presence effect might not be the same for male and female learners, who differ in their perception of social presence. Actually, males and females differ significantly in social brain function when making social decisions from faces [53]. Studies of the instructor presence effect included primarily female participants, who made up around 70% of the sample [8,10,14,19,20,34]. Such a gender imbalance could have skewed the results’ distribution. Therefore, examining the gender differences in online learning, especially in video lectures with an instructor’s presence, is necessary.

### 1.3. The Present Study

As far as we know, our research is the very first to look into gender differences in the instructor presence effect in video lectures. We explored whether men and women differed in attention allocation and learning performance in video lectures with either instructor presence or instructor absence. There are mainly two types of video lectures in the literature: lecture videos with slides and modeling videos in which an instructor provides a step-by-step demonstration of how to perform a task or solve a problem [10,19]. We selected the lecture video, the most common type for learning, and included three different formats of video lectures. The first type is the audio-video presentation (AV), which contains video talks from presentation slides, supplemented with the instructor’s narration without visual presence. This type of video lecture has been widely used for e-learning due to its cost efficiency [54]. The second is the picture-video presentation (PV), which features a teacher’s image (picture) in the presentation slides. The instructor image provides the instructor’s social presence but is less interactive and distractive than the teacher’s talking head [26]. The third condition comprises a synchronized video of the instructor explaining the content and a video of corresponding presentation slides (VV). High media richness is characteristic of videos of this type [6]. The instructor’s image or video was continually displayed in the top-right corner as the default talking head in Zoom meetings. The instructor in the VV condition looked at the camera and spoke naturally without deliberate facial expression or eye gaze, as shown in most online lectures and courses. In each of the three conditions (AV, PV, VV), every participant watched three videos while their eye movement was simultaneously recorded. The comprehension test following each video measured their learning performance.

We hypothesized that the instructor presence effect would be significant in male and female learner groups. As females have been suggested to be more sensitive to social presence than males, they might allocate more visual attention to the instructor than the males. This preference could be displayed in the eye data of fixation and dwell time. Meanwhile, males were less sensitive to social cues, so their attention allocation might be more distributed than the female participants. Since both genders have compensatory online learning strategies, they could achieve similar learning performance.

Therefore, using the eye-tracking technique, the current study investigated the eye movement patterns in male and female students who learned video lectures with/without the instructor’s presence. Our findings should benefit the current understanding of social agency theory regarding learners’ differences and thus improve the effectiveness of online education.

## 2. Materials and Methods

### 2.1. Participants and Design

Sixty-six undergraduates (34 males; age range: 18–21) from a Chinese university participated in the study. All participants had normal or corrected-to-normal vision and hearing. They provided written informed consent before the experiment and were paid for their participation. Two male participants were excluded from the data analysis because they experienced problems in the eye movements’ calibration phase, and 64 participants (32 females; mean age = 19.72 ± 1.02) remained for the data analysis.

This study adopted a two-factor mixed design. Male and female participants were asked to watch nine videos in three instructor conditions (audio-video, AV; picture-video, PV; video-video, VV). They were asked to make true or false judgments about a series of statements following each video. Their scores on this comprehension task indicated their learning performance. All the audio and video stimuli were in Chinese, the participants’ first language. All participants provided informed consent and received payment for their participation. The present study was approved by the ethical committee of the Bilingual Cognition and Development Lab at the Guangdong University of Foreign Studies, China.

### 2.2. Apparatus and Eye Movement Data Analysis

Participants’ eye movement data were collected via an Eyelink 1000 eye tracker (SR Research Ltd., Mississauga, ON, Canada) in the desktop-mounted mode, with a sampling rate of 1000 Hz. Participants were seated approximately 60 cm from the screen. A chin rest was used to minimize their head movements. Each video had three areas of interest (AOI): the text area, the topic-related picture area, and the instructor area. Those instructor-absent videos (the AV condition) did not have an instructor area. We created a corresponding AOI with the equivalent size and location of the instructor AOI from the instructor-present videos [12]. Within each AOI, we collected the participants’ fixation count (average number of total fixations on a particular AOI), fixation count percentage (average percentage of all fixations on a specific AOI), dwell time (average sum of all fixation duration on a specific AOI), dwell time percentage (average percentage of trial time spent on a specific AOI), and number of transitions between different AOIs. All the data were collected at the Bilingual Cognition and Development Lab at the Guangdong University of Foreign Studies. The eye movement data were preprocessed in Data Viewer (SR Research), in which unsuccessful trials (the tracking ratio was lower than 90%) were discarded.

### 2.3. Materials

Eleven video lectures were used in the current study, nine for experimental stimuli and two for the practice session. Those videos introduced topics in science, history, and literature. Details of these videos are presented in Table 1. We downloaded the original passages from the Chinese version of Wikipedia (https://zh.wikipedia.org/wiki/Wikipedia (accessed on 2 May 2021)) and then revised each text into a 400-word script. We asked 20 Chinese students from the same university to rate the familiarity (from 1 not familiar at all to 5 very familiar) and difficulty of its content (from 1 not difficult at all to 5 very difficult) of each topic on two five-point Likert scales. Generally, they reported being not familiar with the topics (mean rating scores = 2.03 ± 1.22) and being not difficult (mean rating scores = 2.23 ± 0.93) with the content of the topics.

Based on those scripts, 11 videos were recorded by the same instructor (a young female native Chinese speaker with the standard accent of Putonghua) in three instructor conditions: audio-video without instructor presence (AV); picture-video with instructor presence (PV); video-video with instructor presence (VV) (Figure 1). Each video lasted about two minutes. In the AV condition, there was only learning content accompanied by the instructor’s narration and a topic-related picture. In the PV condition, a static image of the instructor appeared in the screen’s upper-right corner, with the text and pictures as in the AV condition. In the VV condition, the static image of the instructor was replaced by the instructor’s video giving the talk, as the slides showed. All videos were identical in the size of the text area (454 × 630 pixels), the topic-related picture area (220 × 246 pixels), and the instructor area (260 × 260 pixels). Nine videos were presented in a randomized order for all participants.

### 2.4. Measurements

Comprehension test: After watching each video, participants were instructed to complete eight true or false judgments on visually presented statements based on what they learned from the video. Each question would appear in the same window as the lecture video (Figure 1d). Participants pressed the Yes/No buttons to indicate true or false to those statements. Comprehension scores were calculated by assigning one point for a correct response and zero for incorrect responses. Each participant completed 72 questions for the nine videos (the average accuracy is 85%); the maximum score for their learning performance in each condition was 24. Before the eye-tracking experiment, all the comprehension questions were reviewed and optimized for clarity, accuracy, and content validity by 20 matched control participants who did not join the study.

After the participants finished answering all questions of a video during the eye-tracking experiment, they were asked to rate the familiarity of the topic and the difficulty level of the video content. The instruction clarified that the familiarity referred to their prior knowledge of each topic. As shown in Table 1, participants in the pilot study rated the familiarity and difficulty of the materials; those in the eye-tracking experiment were also unfamiliar with those topics. They all regarded those video contents as moderate to low in difficulty.

### 2.5. Procedure

The participants were tested individually, seated at a desk facing an eye-tracker. Before each video, the participant’s gaze would be calibrated and validated with a 9-point calibration algorithm. Following the eye tracker calibration, participants were given basic instructions and then watched the video. Immediately after watching each video, participants were instructed to answer eight comprehension questions (by pressing the Yes/No buttons on the keyboard). They then rated the familiarity and difficulty of each video (by pressing the keys on the keyboard, from 1 to 5). The nine videos were presented in a randomized order for each participant. They could take a break after watching every three videos. The total duration of this experiment was approximately 40 min.

### 2.6. Data Analysis

We used linear mixed-effects models (LMMs) [55] achieved by the lme4 package in the R environment (version 4.1.0) [56]. In our analyses, we adopted the maximal random-effects structure [57], with *instructor* (AV, PV, and VV) and *gender* (male vs. female) as fixed factors, participants, and video areas (items) as crossed random factors. A random slope would be kept if its inclusion significantly improved the model’s goodness of fit. Besides, as *instructor* was a three-level categorical predictor, we adopted the treatment coding and turned it into two contrasts [58], with the first contrast comparing the AV and PV conditions and the second comparing the AV and VV conditions.

## 3. Results

### 3.1. Effects of Gender and Instructor on Learning Performance

We first examined whether *gender* and *instructor* would exert any influence on learners’ comprehension performance. The LMM results (Table 2) did not reveal a significant effect of *gender* [*β* = 0.01, *SE* = 0.16, *p* = 0.949]. However, we found a significant main effect of the *instructor*. To be more specific, it was found that participants obtained higher scores in the VV condition than in the AV condition [*β* = 0.43, *SE* = 0.14, *p* = 0.002]. There were no significant differences in the comprehension scores between the AV and PV conditions. Thus, the static image of the instructor was not a significant boost to the participants’ learning performance in the video lectures. Additionally, there were no significant interactions between *instructor* contrasts and *gender*, implying that participants achieved the same learning outcomes in all conditions regardless of their gender (Figure 2).

### 3.2. Effects of Gender and Instructor on Visual Attention Allocation

We analyzed the eye movement data to examine gender differences in visual attention allocation during the learning procedure. Each condition involved three AOIs: the text, the topic-related picture, and the instructor. In each AOI, learners’ fixation count (%), dwell time (%), and the number of transitions were gathered. Table 3 presents the descriptive results. The measures, including fixation count, dwell time, and the number of transitions, were further examined across *instructor* conditions and *gender* using linear mixed-effects models. Table 4 presents the LMM results for each measure.

*Learners’ attention to the text AOI.* The significant effects of *instructor* suggest that learners spent less time on the text in the PV and VV conditions than in the AV condition. In other words, adding an instructor (either in a static picture or a video) diverted some of the learners’ attention from the learning content. However, we did not find any effect of *gender* or significant interactions between *instructor* contrasts and *gender*. This suggests that males and females distributed their attention similarly to the text across conditions.

*Learners’ attention to the picture AOI.* The results showed a significant effect of *instructor*. Specifically, it is in the PV condition that learners pay more attention to the topic-related picture. No significant gender differences were found in fixation count and dwell time.

*Learners’ attention to the instructor AOI.* The significant effects of *instructor* suggest that the added instructor in both the PV and VV conditions did attract much of the learners’ attention. Besides, as for dwell time, a significant interaction between *instructor* and *gender* was found. Follow-up tests showed that female learners dwelt longer on the presented instructor than males in the VV condition (*p* < 0.05) (Figure 3).

We also examined the number of transitions between different AOIs, and the direction of transitions was also taken into account. As for the transitions between the text AOI and the instructor AOI, there were significant effects of *instructor*, showing that participants made more transitions between these two AOIs in the PV and VV conditions than in the AV condition. Besides, we also found significant interactions between *instructor* and *gender*. Follow-up tests showed that males transited more between these two AOIs than females (ps < 0.05) (Figure 4). Significant effects of *instructor* for the transitions between the instructor AOI and the picture AOI were also found, indicating that learners transited more between these two AOIs in the PV and VV conditions than in the AV condition. Last, for the transitions between the text AOI and the picture AOI, there were significant effects of *instructor*, showing that learners transited less between these two AOIs in the PV and VV conditions than in the AV condition. In summary, the significant effects of *instructor* generally found for the number of transitions across AOIs suggest that the added instructor in the PV and VV conditions split learners’ attention while they were watching video lectures.

## 4. Discussion

Online learning boomed during the pandemic. The change in learning requires empirical research to verify effective teaching methods in online learning environments. The present study aimed to assess gender differences in the instructor presence effect of video lectures, a central component of the online learning experience. Using the eye-tracking technique, we examined the learning outcome and learning process in male and female Chinese adult learners who took video lectures. In both groups, we found the instructor presence effect: males and females learned better in the videos with the instructor talking than those without the instructor or with the instructor’s static image. In addition, we revealed some gender differences in their attention allocation during the process of video lecture learning.

### 4.1. The Instructor’s Active Engagement in Video Lectures Facilitates Learning Performance in Both Male and Female Adult Learners

An initial objective of this project is to identify the instructor presence effect regardless of gender. We found a significant main effect of *instructor*. The video lectures with the instructor explaining the content of the slides boost the learning performance of male and female learners. Our results support the claim of instructor presence’s facilitation function on online learning performance [6,7,8,9,10,12,18]. This observation supports the hypothesis of the social agency theory [1]: the instructor as a social cue, only in the video condition, might prime a feeling of social presence in learners. The dynamic presence of the instructor might make them more committed to actively processing the provided information and thus improve learning performance.

The instructor’s static presence in the PV condition, though not significantly, benefits the learning of video lectures compared with the baseline condition without the presence of an instructor (Figure 2). This might be explained by the fact that the instructor’s image displayed limited nonverbal cues without mutual eye gaze and active engagement as in the VV condition. For the lack of these embodiment cues, learners might have much less perception of the instructor’s engagement and, therefore, less interaction with the instructor when they just saw the instructor’s picture on the screen. Actually, they did pay some attention (3% dwell time, Table 3) to the instructor’s picture, which is limited compared with that of the VV condition (over 10% dwell time). Our findings suggested that only active instructor engagement in the video lectures is crucial for the facilitation effect of instructor presence.

Our results also shed light on the instructor’s embodiment. The instructor’s static picture represented a low embodiment (without no movement at all). It had been demonstrated ineffective in improving learning outcomes, and the instructor’s video demonstrated a mediate embodiment. Unlike the presented instructor (with a high embodiment) in most previous studies [14,19,32,33,34,35], the instructor in our VV condition did not show any deliberate facial expression, eye gaze, or gestures. It simulated the natural presence of an instructor in some online learning settings, such as the Zoom meeting. In our study, such a natural dynamic presence has also been demonstrated to be effective in learning performance improvement. Therefore, we suggest the teachers turn on their cameras and show their presence in daily online courses.

### 4.2. Males and Females Achieve the Same Performance via Different Attention Allocation Processes

To the best of our knowledge, this is the first study to explore gender differences in the instructor presence effect during video lectures. We compared the learning performance of male and female learners in terms of their comprehension scores after video lectures. We also examined their attention allocation during the learning process using eye-tracking technology. Contrary to expectations, this study did not find a significant gender difference in their learning performance. However, in the VV condition (featuring a video of the instructor), we found some significant gender difference in their attention allocation during the learning process. Male and female learners differed in the dwell time on the instructor AOI, and the number of transitions between the text and the instructor AOIs.

Gender differences, as a personal and fundamental characteristic of learners, have been found in the perception of social presence in web courses [22], online learning strategies [59], and communication efficacy [60]. For example, female learners seemed more sensitive to social presence [23]. They experienced a higher level of social presence during online learning [22]. In the current study, gender differences in attention allocation were found when learners watched the video lectures with the instructor’s video present. Specifically, the results demonstrated that females spent a longer time on the AOI of the instructor. At the same time, males transited more frequently between the instructor and the text. As a social cue to increase social presence, the on-screen instructor made males switch between contents more frequently and did not sustain their attention on the instructor. This might reflect males’ distinct approach to social cues. As a previous study pointed out, men tended to miss social cues and have difficulty processing those cues in social tasks [53]. On the contrary, female learners tended to dwell longer on the instructor. As previous studies demonstrated, this might also be due to females’ sensitivity to the social presence in online learning environments.

Unfortunately, we did not collect data on their perception of the instructor’s social presence, making it hard to explain the findings comprehensively. It is unclear whether females’ attention preference for the instructor indicates their actual liking and interest. Besides, despite the different attention allocation processes, males and females in our study achieved the same learning outcomes after watching the instructor-present videos. Such inconsistent gender differences found in attention allocation and learning outcomes also indicate the necessity of including subjective measures to reveal learners’ actual perceptions. In other words, gender differences in the instructor presence effect should be thoroughly examined regarding learners’ perception, learning outcomes, and attention distribution. Possible relationships among the three dimensions should also be considered when discussing the moderating role of gender in the instructor presence effect.

Last, as mentioned before, the presented instructor in our study did not show salient social cues such as facial expression, body orientation, and gestures. It remains unknown whether males and females would react to the instructor presence differently when presented with an instructor with high embodiment social cues. In that situation, gender differences in learning performance and attention allocation might be more evident. Future research on gender differences could tap into high embodiment settings.

### 4.3. Implications, Limitations, and Future Study

In sum, this study revealed positive effects of instructor presence on participants’ learning outcomes. It highlighted that the positive instructor presence effect held true in male and female learners and suggested including the talking instructor during online lecturing. The learning process regarding eye movement patterns showed some differences for the two groups, with females attending more to the instructor’s video. Future work should measure learners’ perception of social presence to better account for females’ attention preference for the instructor. Finally, our results also caution the problem of imbalanced gender ratio in online learning research and call for the consideration of gender when exploring the effectiveness of online instruction.

This study has a few limitations that future research should consider and address. First, our participants self-reported being generally unfamiliar with the topics covered in this study, but we could have used a prior knowledge test to gauge their pre-existing knowledge of each topic accurately. Besides, we did not use subjective measures to reveal learners’ perception of social presence in video lectures with/without instructor presence. Future research should consider learners’ social presence perception when discussing the potential gender differences in the instructor presence effect. Finally, this study only tested participants’ retention for assessment of learning. It is better to include a transfer test to see how well learners understand the material [21]. Therefore, it still remains an open question whether males and females will differ in transferring what they have learned from video lectures with (or without) the instructor’s presence. Future research on gender differences in the instructor presence effect should adopt retention and transfer tests to assess learning outcomes.

## 5. Conclusions

In response to the call for more attention on individual differences in instructor presence research, the current study focused on gender differences. Using eye-tracking technology, we examined male and female learners’ attention allocation and learning outcomes in video lectures with the instructor present or absent. The instructor talking video facilitated learning performance in both genders, who achieved similar learning performance. However, the male and female learners showed different attention allocation patterns: females dwelt longer on the talking instructor, and males switched more between the instructor and learning content. Future research should comprehensively explore the potential gender differences in the instructor presence effect by examining learners’ perception, learning outcome, and attention allocation in high embodiment settings.

## Figures and Tables

**Figure 1 brainsci-12-00946-f001:**
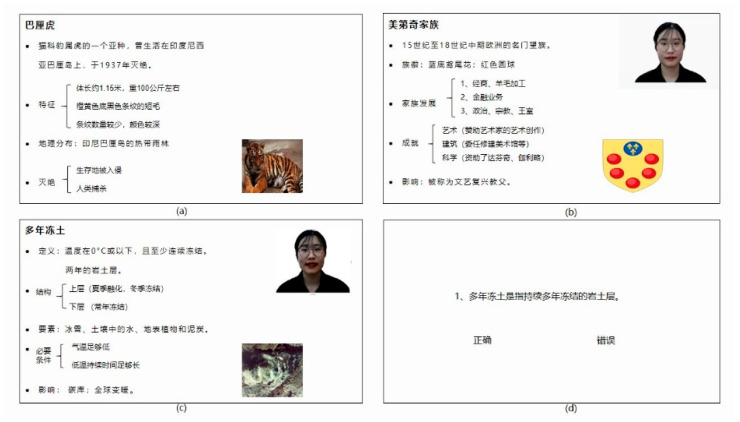
Screenshots of the videos in the (**a**) audio-video condition (AV), (**b**) picture-video condition (PV), (**c**) video-video condition (VV), and (**d**) one comprehension question following a video.

**Figure 2 brainsci-12-00946-f002:**
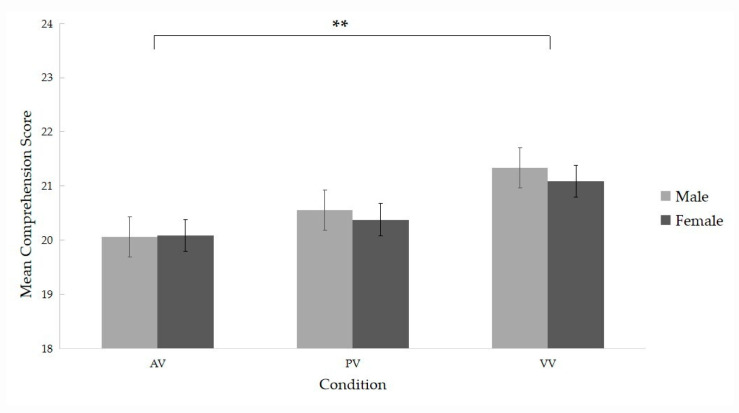
Leaning performance of male and female learners in three instructor conditions. AV, the audio-video condition; PV, the picture-video condition; VV, the video-video condition. ** *p* < 0.01.

**Figure 3 brainsci-12-00946-f003:**
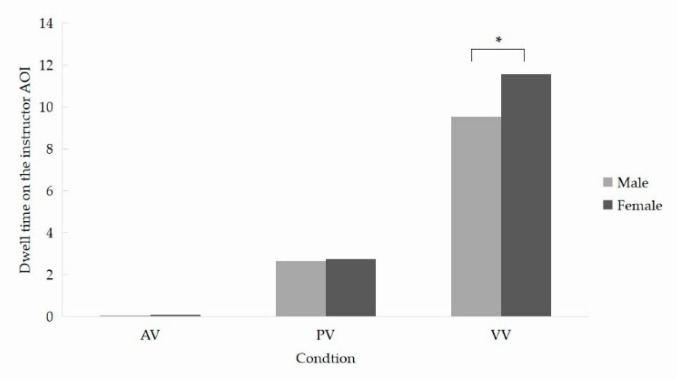
Learners’ dwell time on the instructor AOI. AV, the audio-video condition; PV, the picture-video condition; VV, the video-video condition. * *p* < 0.05.

**Figure 4 brainsci-12-00946-f004:**
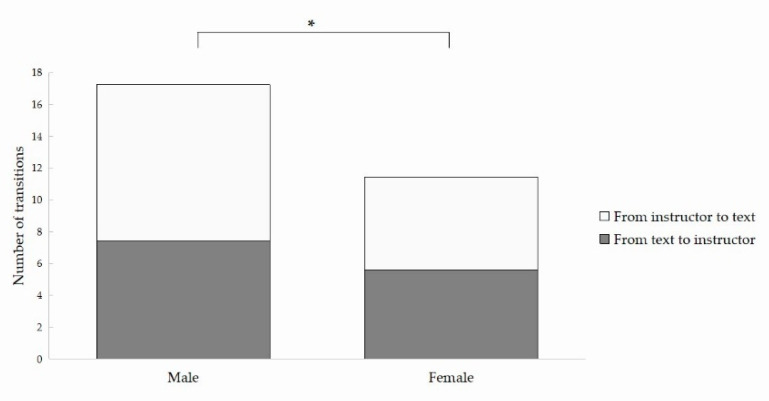
Number of transitions between the text and the instructor in the VV condition. * *p* < 0.05.

**Table 1 brainsci-12-00946-t001:** Descriptives of 11 lecture videos.

NO.	Topic	Area	Condition	Familiarity ^a^	Difficulty ^a^	Familiarity ^b^	Difficulty ^b^
1 *	Venus	Science	PV	3.21	2.47	2.29	2.70
2 *	Volcano	Science	VV	3.63	2.47	2.92	2.65
3	Rosetta stone	History	VV	1.37	2.16	1.38	2.94
4	Medici	History	PV	1.95	2.05	1.91	2.18
5	Copper age	History	AV	1.58	2.63	1.56	2.59
6	The sound and the fury	Literature	PV	2.42	2.26	1.56	3.06
7	Isabel Allende	Literature	AV	1.42	2.05	1.58	2.41
8	Malin Kundang ^c^	Literature	VV	1.26	1.58	1.42	2.00
9	Rhizanthella gardneri ^d^	Science	PV	1.31	2.15	1.50	2.56
10	Balinese tiger	Science	AV	1.95	1.79	1.76	2.3
11	Permafrost	Science	VV	2.16	2.42	2.53	2.67

Note. AV, the audio-video condition; PV, the picture-video condition; VV, the video-video condition; * used as practice; ^a^ rated by 20 Chinese students who did not participate in the study; ^b^ rated by the 64 participants in the eye-tracking experiment; ^c^ a folk tale in Southeast Asia; ^d^ an entirely subterranean mycoheterotrophic orchid.

**Table 2 brainsci-12-00946-t002:** Results of linear mixed-effects models for comprehension scores.

Effect	*β*	*SE*	*t*	*p*
Intercept	6.69	0.19	36.05	<0.001 ***
Instructor 1: AV vs. PV	0.16	0.14	1.14	0.257
Instructor 2: AV vs. VV	0.43	0.14	3.10	0.002 **
Gender	0.01	0.16	0.06	0.949
Instructor 1: Gender	−0.06	0.19	−0.32	0.748
Instructor 2: Gender	−0.08	0.19	−0.428	0.669

Note. The final LMM included both by-participant and by-item intercepts. ** *p* < 0.01; *** *p* < 0.001.

**Table 3 brainsci-12-00946-t003:** Visual attention distribution statistics for the videos in the AV, PV, and VV conditions.

AOI	Measure	AV		PV		VV	
		Male	Female	Male	Female	Male	Female
Text	Fixation count	244.90 (30.60)	257.67 (44.81)	217.96 (30.17)	221.92 (45.63)	211.33 (35.14)	221.41 (49.45)
	Fixation count (%)	87.61 (0.04)	87.42 (0.07)	80.62 (0.06)	80.54 (0.08)	78.37 (0.08)	79.14 (0.10)
	Dwell time ^a^	77.47 (8.51)	79.82 (11.64)	70.55 (9.43)	71.44 (11.54)	66.94 (11.10)	67.48 (14.90)
	Dwell time (%)	87.14 (0.05)	86.59 (0.08)	80.69 (0.07)	79.65 (0.10)	75.80 (0.11)	74.86 (0.14)
Picture	Fixation count	30.64 (11.64)	32.56 (18.5)	38.83 (17.80)	39.82 (21.22)	29.74 (15.74)	30.18 (17.78)
	Fixation count (%)	10.84 (0.04)	11.03 (0.06)	14.16 (0.06)	14.37 (0.07)	10.98 (0.06)	10.67 (0.06)
	Dwell time	10.46 (4.86)	10.97 (6.41)	12.99 (6.76)	13.98 (6.81)	10.38 (6.54)	9.89 (5.68)
	Dwell time (%)	11.66 (0.05)	12.17 (0.07)	14.86 (0.07)	15.76 (0.08)	11.60 (0.07)	11.30 (0.07)
Instructor	Fixation count	0.25 (0.63)	0.29 (0.92)	8.85 (6.18)	8.83 (7.95)	23.65 (19.88)	23.82 (20.02)
	Fixation count (%)	0.09 (0.20)	0.11 (0.44)	3.20 (0.02)	3.21 (0.03)	8.53 (0.07)	8.79 (0.07)
	Dwell time	0.07 (0.18)	0.09 (0.39)	2.67 (1.95)	2.75 (2.47)	9.55 (7.28)	11.58 (10.58)
	Dwell time (%)	0.08 (0.20)	0.10 (0.45)	3.07 (0.02)	3.13 (0.03)	10.91 (0.08)	12.83 (0.11)

Note. AOI, area of interest; AV, the audio-video condition; PV, the picture-video condition; VV, the video-video condition; ^a^ The unit of dwell time was second.

**Table 4 brainsci-12-00946-t004:** LMM statistics for eye movement measures.

		Instructor						Gender			Interaction					
		Instructor 1: AV vs. PV			Instructor 2: AV vs. VV						Instructor 1: Gender			Instructor 2: Gender		
AOI	Measure	*β*	*SE*	*p*	*β*	*SE*	*p*	*β*	*SE*	*p*	*β*	*SE*	*p*	*β*	*SE*	*p*
Text	Fixation count ^a^	−38.04	5.58	<0.001 ***	−39.43	5.58	<0.000 ***	−16.17	10.34	0.121	12.02	7.89	0.129	5.85	7.89	0.459
	Dwell time ^a^	−9.42	1.69	<0.001 ***	−13.09	1.69	<0.001 ***	−3.32	2.93	0.259	2.39	2.39	0.317	2.47	2.39	0.303
Picture	Fixation count ^a^	6.23	2.65	0.019 *	−3.01	2.65	0.256	−2.14	4.36	0.625	2.12	3.74	0.572	2.15	3.75	0.567
	Dwell time ^a^	2.62	1.02	0.011 *	−1.31	1.02	0.199	−0.55	1.58	0.727	−0.15	1.44	0.919	1.20	1.44	0.404
Instructor	Fixation count ^b^	8.19	1.59	<0.001 ***	23.39	1.59	<0.001 ***	−0.04	2.43	0.986	0.39	2.26	0.861	−1.04	2.26	0.645
	Dwell time ^b^	2.59	0.76	<0.001 ***	11.41	0.76	<0.001 ***	−0.02	1.07	0.983	8.19	1.07	0.993	−2.23	1.07	0.038
Number of transitions																
	Instructor→Text ^b^	3.03	1.39	0.031 *	5.64	1.39	<0.001 ***	−0.02	1.53	0.989	0.21	1.98	0.916	4.05	1.98	0.041 *
	Text→Instructor ^b^	2.39	0.48	<0.001 ***	5.50	0.48	<0.001 ***	<.001	0.75	1.000	0.39	0.67	0.569	1.77	0.67	0.009 **
	Picture→Instructor ^b^	1.19	0.17	<0.001 ***	1.56	0.17	<0.001 ***	0.02	0.23	0.931	−0.06	0.25	0.800	0.16	0.25	0.518
	Instructor→Picture ^b^	0.77	0.18	<0.001 ***	1.57	0.18	<0.001 ***	<.001	0.23	1.000	0.39	0.26	0.129	0.38	0.26	0.151
	Picture→Text ^c^	−1.69	0.75	0.027 *	−3.32	0.82	<0.001 ***	0.56	1.39	0.689	1.48	1.06	0.166	−0.73	1.15	0.529
	Text→Picture ^a^	−0.98	0.63	0.120	−2.98	0.63	<0.001 ***	0.79	1.19	0.508	0.69	0.88	0.433	−0.98	0.89	0.271

Note: ^a^ The final LMM included both by-participant and by-item intercepts; ^b^ the final LMM included a by-participant intercept; ^c^ the final LMM model included a by-participant random slope for condition, in addition to by-participant and by-item intercepts. AOI, area of interest; AV, the audio-video condition; PV, the picture-video condition; VV, the video-video condition. * *p* < 0.05, ** *p* < 0.01, *** *p* < 0.001.

## Data Availability

The data presented in this study are available on request from the corresponding author. The data are not publicly available due to privacy and ethical restrictions.

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
