# Peer review of "Exploring Gender Differences in the Instructor Presence Effect in Video Lectures: An Eye-Tracking Study"

_brainsci, 2022, doi:10.3390/brainsci12070946_

Round 1
Reviewer 1 Report
I have read the manuscript entitled ‘Exploring gender differences in the instructor presence effect during video lectures: An eye-tracking study’ with great interest. The study reports on an experiment in which students were asked to watch thee videos under different conditions: no instructor present, instructor present (picture), instructor present (video). Both learning outcomes (immediate posttest with true/false questions) and attention allocation were examined in relation to the three different conditions and gender differences. The study did not find any statistically significant differences between male and female students.
I have identified some issues in the manuscript that require revision and I will outline them one by one below. In its current form I cannot recommend the article for publication.
Introduction
- In the second paragraph of the introduction the authors claim that most studies focus on students’ learning outcomes when examining the role of instructor presence. I do not agree with this claim since there is growing empirical research focusing on attention allocation as measures with eye-tracking in relation to instructor presence. The manuscript would benefit from an elaborate introduction on these studies as well and how differences in attention allocation have been captured in previous work. This part is largely missing from the introduction and theoretical framework. I highly recommend the authors to include this work in both the introduction and theoretical framework as well. Claiming that most research focuses on learning outcomes is not true. I can see a lot of relevant papers in the reference list on the topic of attention allocation and instructor presence, but an in-depth discussion of this work is missing and needs to be elaborated on (line42-43).
- In the last sentence of the introduction the authors indicate that their findings can help verify gender equality. I do not fully agree here: the main question this study can help answering is whether there are gender differences in processing video lectures which is not the same as gender equality in education (line 59-60).
- The social agency theory is introduced in paragraph 1.1.1. However, I miss the information on how the instructor can use different cues while being present to signal important information in the video which is an important multimedia principle derived from the cognitive load theory and theory of multimedia learning. There is a huge differences between an instructor staring at the camera the whole time (huge interference) and an instructor using social cues such as gaze and gesture guidance (less interference). This paragraph would benefit from a more nuanced view on both theories and how instructor presence might reduce or increase cognitive load as well.
- When the instructor presence effect is discussed a distinction should been made between modelling videos and lecture videos. Also here instructor presence might play a different role.
- The part on eye-tracking needs elaboration (1.3). There are only a limited number of studies described here (see also my first point). In addition, it is not well described what how the different eye-tracking measures need to be interpreted in relation to the introduced theories.
Methods
- Paragraph 2.1: Were there any other quality checks carried out to check data quality?
- Paragraph 2.1: The PV condition is not very interesting in my opinion since a picture of an instructor cannot signal important information in the video. However, this is something that cannot be changed at this point.
- Paragraph 2.3: Can the authors clarify how much videos students watched? Did they see all nine videos?
- Paragraph 2.3: I am a bit puzzled by the huge difference in topics used for the videos going from science to history and literature. Students might have different interests for these topics and this might interfere with the conditions the authors want to test. Moreover, the effect of topic was not controlled for in the analysis and might have impacted the results to a large extent. The area of the videos could be taken into account in the analysis when using mixed effects models. I would like to convince the authors that with the data at hand, mixed effects models would be the better way to analyse the data and it would also allow to take the area level into account. Mixed effects models would allow to take into account random effects for the participants and area of the video. By doing so, one does not need to average across video areas and one can take into account the variation caused by area as well. In my opinion, this would be the right approach to analyse the data and it might have a huge impact on the results. I highly recommend the authors to re-analyse the data using mixed effect models.
More information on mixed effects models can be found here:
Baayen, R. H., Davidson, D. J., & Bates, D. M. (2008). Mixed-effects modeling with crossed random effects for subjects and items. Journal of Memory and Language, 59(4), 390-412. https://doi.org/10.1016/j.jml.2007.12.005.
Reviewer 2 Report
The paper explores gender differences about the presence effect of instructors in video lectures through eye-tracking study. In agreement with prior literature, the authors found that students performed better with instructors’ dynamic videos than without video or with still image of the instructor. In terms of gender, the authors did not find any difference between male and female students’ learning outcomes and eye-movements. I have a few suggestions.
I would advise the authors to further strengthen their arguments on how learning (I think it should be learning outcomes) can be improved from instructor’s dynamic presence through increased social presence and social agency. There are a couple of references but the link is not so clear.
In the paper, it would be great that rather than claiming that students’ learning is improved, the authors update it to reflect that in fact it’s the learning outcomes or scores that are improved. I am suggesting this as learning is not a measure of scores, it is growth that should be observed over time.
On lines 57-58: ‘they learned video lectures’. Did you mean: they learned through video lectures?
On lines 73-74: ‘divert learners’ limited attention to the learning content’. Did you mean: divert from the learning content?
On page 3, lines 122-130, some of the references are rather old, it would be appreciated if more new references could be included.
Does the last paragraph on page 3 represent your research hypothesis? If so, I would suggest making it more prominent for the reader.
In Section 2.2, while considering the transitions between instructor presence and AOIs did you consider the complexity of the learning content? I presume, the transition rate would be higher towards AOIs when the content is more complex and vice-versa for the easier content. Similarly, if the slides are more content-heavy that can also affect the transition rate and the subjective difficulty and hence student attention. While the authors mention that “males and females have distributed their attention similarly while watching the videos without the instructor, with more than 85ï¼… of the time allocated to the text”, I personally feel that this should be inspected further in terms of gaze direction.
In Section 2.4, as students’ knowledge or understanding is based on True / False questions, there is a possibility (50% chance per question) that students make a guess. While there were 72 questions, I would still be cautious to infer students’ knowledge or understanding from simple True/False statements. Usage of some reflective questions should also be done to better infer students’ understanding.
I would be happy to accept this publication if the authors could slightly update their methodology and in addition to comparing eye fixations in three broad AOIs (Text, Content Image and Instructor image), they could also compare male and female learners in their gaze direction whether it is forward or behind to that of instructor's spoken words.
